# Development and Evaluation of a Set of Spike and Receptor Binding Domain-Based Enzyme-Linked Immunosorbent Assays for SARS-CoV-2 Serological Testing

**DOI:** 10.3390/diagnostics11081506

**Published:** 2021-08-20

**Authors:** Rosa Camacho-Sandoval, Alejandro Nieto-Patlán, Gregorio Carballo-Uicab, Alejandra Montes-Luna, María C. Jiménez-Martínez, Luis Vallejo-Castillo, Edith González-González, Hugo Iván Arrieta-Oliva, Keyla Gómez-Castellano, Omar U. Guzmán-Bringas, María Pilar Cruz-Domínguez, Gabriela Medina, Laura A. Montiel-Cervantes, Maricela Gordillo-Marín, Roberto Vázquez-Campuzano, Belem Torres-Longoria, Irma López-Martínez, Sonia M. Pérez-Tapia, Juan Carlos Almagro

**Affiliations:** 1Unidad de Desarrollo e Investigación en Bioprocesos (UDIBI), Escuela Nacional de Ciencias Biológicas, Instituto Politécnico Nacional, Mexico City 11340, Mexico; rosa.camacho@udibi.com.mx (R.C.-S.); alejandro.nieto@udibi.com.mx (A.N.-P.); gregorio.carballo@udibi.com.mx (G.C.-U.); alejandra.montes@udibi.com.mx (A.M.-L.); luis.vallejo@udibi.com.mx (L.V.-C.); edith.gonzalez@udibi.com.mx (E.G.-G.); ivan.arrieta@udibi.com.mx (H.I.A.-O.); keyla.gomez@udibi.com.mx (K.G.-C.); uriel.guzman@udibi.com.mx (O.U.G.-B.); 2Laboratorio Nacional para Servicios Especializados de Investigación, Desarrollo e Innovación (I+D+i) para Farmoquímicos y Biotecnológicos, LANSEIDI-FarBiotec-CONACyT, Mexico City 11340, Mexico; 3Departamento de Bioquímica, Facultad de Medicina UNAM, Mexico City 06800, Mexico; mcjimenezm@institutodeoftalmologia.org; 4Departamento de Inmunología, Unidad Periférica “Conde de Valenciana” UNAM, Mexico City 06800, Mexico; 5División de Investigación, Hospital de Especialidades Centro Médico Nacional La Raza, IMSS, Mexico City 02990, Mexico; drapilarcd@gmail.com; 6Unidad de Investigación Traslacional en Enfermedades Hemato-Oncológicas, Hospital de Especialidades Centro Médico Nacional La Raza, IMSS, Mexico City 02990, Mexico; dragabymedina@yahoo.com.mx (G.M.); lauramontielcervantes@outlook.com (L.A.M.-C.); 7Departamento de Morfología, Escuela Nacional de Ciencias Biológicas, Instituto Politécnico Nacional (ENCB-IPN), Mexico City 11340, Mexico; 8Instituto de Diagnóstico y Referencia Epidemiológicos (InDRE) “Dr. Manuel Martínez Báez”, Mexico City 01480, Mexico; maricela.gordillo@salud.gob.mx (M.G.-M.); roberto.vazquez@salud.gob.mx (R.V.-C.); belem.torres@salud.gob.mx (B.T.-L.); irma.lopez@salud.gob.mx (I.L.-M.); 9Departamento de Inmunología, Escuela Nacional de Ciencias Biológicas, Instituto Politécnico Nacional (ENCB-IPN), Mexico City 11340, Mexico; 10GlobalBio, Inc., 320 Concord Ave, Cambridge, MA 02138, USA

**Keywords:** COVID-19, IgG isotypes, serological diagnostics, seroconversion, UDITEST-V2G^®^

## Abstract

The implementation and validation of anti-SARS-CoV-2 IgG serological assays are reported in this paper. S1 and RBD proteins were used to coat ELISA plates, and several secondary antibodies served as reporters. The assays were initially validated with 50 RT-PCR positive COVID-19 sera, which showed high IgG titers of mainly IgG1 isotype, followed by IgG3. Low or no IgG2 and IgG4 titers were detected. Then, the RBD/IgG assay was further validated with 887 serum samples from RT-PCR positive COVID-19 individuals collected at different times, including 7, 14, 21, and 40 days after the onset of symptoms. Most of the sera were IgG positive at day 40, with seroconversion happening after 14–21 days. A third party conducted an additional performance test of the RBD/IgG assay with 406 sera, including 149 RT-PCR positive COVID-19 samples, 229 RT-PCR negative COVID-19 individuals, and 28 sera from individuals with other viral infections not related to SARS-CoV-2. The sensitivity of the assay was 99.33%, with a specificity of 97.82%. All the sera collected from individuals with infectious diseases other than COVID-19 were negative. Given the robustness of this RBD/IgG assay, it received approval from the sanitary authority in Mexico (COFEPRIS) for production and commercialization under the name UDISTEST-V2G^®^.

## 1. Introduction

Coronavirus Disease 2019 (COVID-19) is a viral infection caused by the severe acute respiratory syndrome coronavirus 2 (SARS-CoV-2). COVID-19 was first recorded in December 2019 in Wuhan, China [1,2], and quickly spread globally. It reached Mexico on 27 February 2020 [3], and by the time of writing this report (13 August 2021), the number of COVID-19 confirmed cases in Mexico was close to 3.1 million with 248,000 fatalities (https://covid19.who.int/region/amro/country/mx accessed on 18 August 2021) [4] and a devastating economic impact.

While several vaccines have received emergency approval in diverse countries to prevent COVID-19 [5] and have been widely applied to control further spread of this viral infection, new SARS-CoV-2 variants of concern (VoC) [6,7] are emerging. Several VoCs partially render vaccines ineffective [8], and several are reinfecting recovered patients [9,10]. Thus, COVID-19 diagnosis, including reverse transcription-polymerase chain reaction (RT-PCR) methods and IgG/IgM serological tests to detect anti-SARS-CoV-2 antibodies, continue to be indispensable tools to identify new COVID-19 outbreaks and thus, assist in the management and eventual control of the COVID-19 pandemic as well as ascertain the success of COVID-19 vaccination campaigns.

SARS-CoV-2 is a positive-stranded RNA virus from the beta-coronavirus genera [2]. Its genome encodes for two open reading frames (ORFs) containing the four major structural proteins of the virus: nucleocapsid (N), membrane (M), envelope (E), and spike (S) [11,12]. The N, M, and E proteins are necessary for virus assembly. The S protein is located on the surface of the viral particles and has two subunits. The subunit S1 mediates the attachment, via the receptor-binding domain (RBD), to angiotensin-converting enzyme 2 (ACE2) in the host cells [12], whilst subunit S2 mediates cell fusion [13]. S1 and RBD are highly immunogenic and have been extensively utilized as capture reagents for anti-SARS-CoV-2 IgG serodiagnosis [14,15,16,17,18]. Nevertheless, for the most part, current IgG serological tests based on S1 and RBD are rapid qualitative tests and are not flexible enough to study different components of the antibody response to SARS-CoV-2 infection, for example, IgG isotypes, anti-SARS-CoV-2 antibody titers, and the immunological competence of a vaccinated individual. To bridge this gap, here, we describe the implementation and application of a set of anti-SARS-CoV-2 IgG serological assays based on commercial S1 and in-house produced recombinant RBD as capture reagents.

We report first the optimization of the S1 and RBD assays and definition of positive and negative cut-off values with 15 well-characterized sera from RT-PCR positive sera and 15 negative COVID-19 serum samples. Second, the assays performance with 50 serum samples from COVID-19 RT-PCR positive individuals is reported This initial performance test included assessing the S1/RBD IgG titers, and contribution of IgG isotypes to the antibody response of this panel of serum samples. Third, we present the results of an additional validation of the RBD/IgG assay showing how the assay responded to increases in the anti-SARS-CoV-2 IgG titers as the immune response to COVID-19 infection progressed. This test was performed with another 269 sera from COVID-19 RT-PCR positive individuals collected at 7, 14, 21, and 40 days after the onset of COVID-19 symptoms. Finally, with report and discus a study in collaboration with the Instituto de Diagnóstico y Referencia Epidemiologicos (InDRE), which is the reference organization in Mexico for the diagnostics of COVID-19, in which the RBD/IgG assay performance was evaluated with a panel of 406 sera, comprising 149 COVID-19 RT-PCR positive, 229 COVID-19 RT-PCR negative, and 28 sera collected from individuals with other viral infections (not SARS-CoV2). The RBD/IgG assay performed well in all the studies mentioned above, leading to emergency use authorization by the regulatory agency in Mexico (COFEPRIS) to produce and commercialize the test under the name UDISTEST-V2G^®^.

## 2. Materials and Methods

### 2.1. RT-PCR Assay

Molecular detection of SARS-CoV-2 in nasopharyngeal swabs was conducted using the protocol approved by the World Health Organization (https://www.who.int/docs/default-source/coronaviruse/whoinhouseassays.pdf accessed on 18 August 2021) [19]. The RT-PCR from isolated viral RNA was performed following the protocol published by Corman and co-workers [20], with slight modifications. In brief, gene E, using RNA-dependent RNA polymerase (RdRp), was amplified as a specific target to detect SARS-CoV-2. The presence of Ribonuclease P (RNase P) was used as a control of the RT-PCR. The RT-PCR reaction was performed with the SuperScript III enzyme, following the manufacturer’s recommendations (ThermoFisher Scientific, Waltham, MA, USA; Cat. No. 18080093), in a 7500 Fast Thermocycler (Applied Biosystems, Waltham, MA, USA). The diagnosis was based on the value of cycle threshold (CT) for gene E, where RdRp and RNase *p* values of ≤38 CT were considered positive.

### 2.2. RBD Expression, Purification, and Characterization

The vector and RBD nucleotide sequence used for RBD expression has been published elsewhere and were kindly donated by Dr Florian Kammer at the Department of Microbiology, Icahn School of Medicine, at Mount Sinai, New York, NY, USA. The plasmid DNA was expanded in the *Escherichia coli* DH5α strain, purified using the *EndoFree Plasmid Maxi Kit* (QIAGEN, Hilden, Germany; Cat. No. 12362) and transfected into human embryonic kidney (HEK) 293 cells (ATCC CRL-3216). After a four-day incubation at 5% CO_2_ at 37 °C, the culture supernatant was harvested and filtrated by a 0.45 µM membrane (Millipore, Burlington, MA, USA; Cat. No. HAWG047S6).

The RBD was purified by Immobilized Metal Affinity Chromatography (IMAC), employing an ÄKTA Fast Protein Liquid Chromatography system (Amersham Biosciences, Amersham, UK). The protein was captured on a 5 mL HisTrap™ Nickel column (G.E. Healthcare, Chicago, IL, USA; Cat. No. 17-5255-01), previously equilibrated with a PBS 1X (Gibco; Cat. No. 10010023) + 25 mM imidazole (Sigma-Aldrich, San Luis, MO, USA; Cat. No. I2399-100G) solution at 2.5 mL/min. Low-affinity proteins bound to the column were removed using the same buffer solution at 5 mL/min for 7 min. High-affinity proteins were eluted using a PBS 1× + 250 mM imidazole solution at 5 mL/min. Eluted fractions were collected in 15 mL tubes, and their volume was reduced using a 10 kDa centrifugal filter (Millipore, Cat. No. UFC801096) at 4000× *g* and 20 °C for 12 min. Imidazole was removed from RBD fractions by adding 15 volumes of PBS1X and using the ultrafiltration procedure previously described.

The protein content was quantitated by UV/VIS spectrophotometry on an Epoch System (Bio Tek Instruments, Winooski, VT, USA), using the extinction coefficient 33,850 M^−1^ cm^−1^. Quality control of the RBD protein also included: (i) determination of mass distribution by native size exclusion chromatography (SEC) and (ii) determination of the structural integrity by denaturing SDS-PAGE. SEC analysis was performed on an Acquity^®^ Class-H UPLC system (Waters Corporation, Milford, MA, USA) using a 4.6 × 150 mm BEH 200 column (Waters, Cat. No. 176003905). The sample was eluted using a 6.81 pH phosphate (50 mM)/sodium chloride (150 mM) buffer solution at 0.4 mL/min and 30 °C. Data were acquired using a UV detector at 280 nm and processed using the software Empower (Waters Corporation). The molecular size of the samples was calculated according to the elution profile of a Gel Filtration Standard (Bio-Rad, Hercules, CA, USA; Cat. No. 1511901). Conversely, SDS-PAGE was performed using Any kD™ TGX Stain-Free™ Protein Gels (Bio-Rad; Cat. No. 4568123) and a Mini-PROTEAN^®^ system (Bio-Rad). RBD samples (1 μg/lane) were analyzed under non-reducing and reducing conditions, employing 2-Mercaptoethanol (BME) at 2.5% (Bio-Rad; Cat. No. 1610710) or 1,4-Dithiothreitol (DTT) at 5 mM (Sigma-Aldrich). The reduction was achieved by incubating the samples at 95 °C for 15 min. The data were acquired using the ChemiDoc Imaging Systems (Bio-Rad).

### 2.3. S1- and RBD-Based IgG Assays

After exploring several coating and incubation conditions as well as diverse reagent concentrations to maximize the difference between the panel of negative and positive serum samples used for the optimization of the assays (see below), the following protocol was established. Fifty µL of S1 (Sino Biological, Beijin, China; Cat. No. 40591-V08H) or the RDB produced in house at 1 µg/mL in PBS (Gibco, Amarillo, Texas; Cat. No. 70011-044) were used to coat Nunc MaxiSorp™ flat-bottom plates (ThermoFisher Scientific; Cat No. 44-2404-21). After an overnight incubation at 4 °C, the plates were washed three times with PBS Tween-20 0.1% (PBS-T) and blocked with 200 µL of skim milk 3% prepared in PBS Tween-20 0.1% (MPBS-T 3%) for one hour at room temperature (R.T.). Afterward, the plates were washed three times with PBS Tween-20 0.1%, the content of the wells was removed, and the plates were stored until use.

Following Amanat and co-workers recommendation, all the samples were incubated at 56 °C for one hour prior to the assays to inactivate potential SARS-CoV-2 circulating in the blood at the time of the sample collection. The samples were then diluted at 1:100 in MPBS-T 1% for S1 and MPBS-T 3% for RBD to perform IgG detection. One hundred µL of each dilution was loaded in the S1 or RBD coated plates and incubated at R.T. for 1.5 h. Next, the plates were washed six times with PBS-T 0.1%. S1- and RBD-bound antibodies were detected with 50 µL of diverse secondary antibodies (see below). The assays were completed with the TMB Substrate Reagent Set (B.D. OptEIA, BD Biosciences, Franklin Lakes, NJ, USA; Cat. No. 555214), and the reaction was stopped after 20 min by the addition of methanesulphonic acid (abcam; Cat. No. ab171529). The absorbance was read at 450/570 nm, using an automated SpectraMax M3 Multi-Mode Microplate reader (Molecular Devices, San Jose, CA, USA).

### 2.4. Optimization Set of 15 Positive and 15 Negative Serum Samples

Proof of concept and optimization of the S1/RBD IgG assay conditions were performed with 15 positive and 15 negative serum samples. The positive sera were obtained from confirmed RT-PCR positive patient from Centro Médico La Raza, Instituto Mexicano del Seguro Social (IMSS), in Mexico City. These individuals also exhibited the typical symptoms of a COVID-19 infection at the time of the RT-PCR test. The serum samples of these patients were collected at least 25 days after COVID-19 RT-PCR positive diagnostics. The 15 negative serum samples came from two sources: (1) 6 serum samples were collected in 2018, before the COVID-19 pandemic, and (2) 9 samples were collected from colleagues who had been monitored by RT-PCR since the onset of the pandemic and had consistently given negative results.

### 2.5. Initial Validation Panel of 50 COVID-19 RT-PCR Positive Sera

The 50 serum samples used as the initial validation set of the S1 and RBD assays were obtained from three sources and were different from the 15 positive serum samples used in the assay optimization. The first source (*n* = 21) was health care workers heavily exposed to SARS-CoV-2 at Centro Médico La Raza, Instituto Mexicano del Seguro Social (IMSS), in Mexico City. These serum samples were collected 15 or more days after the onset of COVID-19 symptoms. The second source (*n* = 17) was patients not involved in healthcare services who were diagnosed as COVID-19 positive by RT-PCR, and whose serum samples were donated to be included in this and other studies. Eleven of these patients were confirmed as COVID-19 positive by RT-PCR in our laboratory, and six were diagnosed elsewhere. All these individuals were diagnosed as COVID-19 positive 30–52 days before the serum sample collection. The third source (*n* = 12) was participants enrolled in an approved study at Instituto Politécnico Nacional (IPN) who exhibited the typical symptoms of COVID-19, and voluntarily participated in this study. Samples were collected with written informed consent with approval protocol authorization R-2020-3501-108 (approval day 10 June 2020).

### 2.6. Secondary Antibodies and Isotyping

The S1/RBD total IgG assays were revealed with 50 µL of goat anti-human (IgG Fc) secondary antibody horseradish peroxidase (HRP) conjugated (1:15,000, Abcam, Cambridge, UK; Cat. No. Ab97225). The contribution of different IgG isotypes to the IgG response was assessed with 50 µL of anti-human IgG1 Fc-HRP (Southern Biotech, Birmingham, AL, USA; Cat. No. 9054-05) (1:3000), anti-human IgG2 Fc-HRP (Southern Biotech; Cat. No. 9060-05) (1:3000), anti-human IgG3 hinge-HRP (Southern Biotech; Cat. No. 9210-05) (1:3000), and anti-human IgG4 Fc-HRP (Southern Biotech; Cat. No. 9200-05) (1:3000).

### 2.7. Extended Validation with a Panel of COVID-19 Serum Samples Collected at Different Times

The RBD-IgG assay, once optimized and validated, was assembled as a ready-to-use kit called UDITEST-V2G^®^ and tested with an expanded panel of serum samples. The serum samples were collected from individuals with COVID-19 RT-PCR positive results, who referred COVID-19 symptoms. An initial set of 269 sera were collected on day 7 after the onset of COVID-19 symptoms. Out of this initial group, 214 individuals also donated sera on day 14, and 206 individuals further donated serum samples on day 21. Finally, 198 out of the initial 269 patients donated serum samples on day 40. All 887 serum samples were processed with UDITEST-V2G^®^, as described in previous sections.

### 2.8. Performance of UDITEST-V2G^®^ at InDRE

To assess the sensitivity and specificity of UDITEST-V2G^®^ by an independent laboratory, we established a collaboration with InDRE. They used a panel of 149 serum samples from patients confirmed COVID-19 RT-PCR positive, who had been referred due to symptoms of COVID-19, or individuals with confirmed COVID-19 RT-PCR positive who were in direct contact with a COVID-19 patient. InDRE also tested a panel of 229 negative serum samples with UDITEST-V2G^®^, including 121 COVID-19 RT-PCR negative samples, plus 108 serum samples collected before the COVID-19 pandemic. Further, InDRE used a panel of 28 serum samples from individuals negative to COVID-19, but with diagnoses of other viral infections to determine the specificity of UDITEST-V2G^®^. UDITEST-V2G^®^ was performed at InDRE’s facilities by its personnel, following the UDITEST-V2G^®^ protocol described in previous sections.

## 3. Results

### 3.1. RBD Expression, Purification, and Characterization

The expression yield of the RBD in HEK 293 cells four days after transfection was approximately 10 mg/L of culture after purification. Two fractions were obtained from the RBD purification by IMAC: Fraction 1 (F1) evinced one ~100 kDa main peak (70.1%), and Fraction 2 (F2) showed one ~30 kDa main peak (82.4%) as presented in Figure 1A. Both RBD fractions showed activity with enzyme-linked immunosorbent assay (ELISA, data not shown), indicating the presence of monomeric RDB (~30 kDa) and multimeric (trimer) RBD (~100 kDa) in the obtained fractions. Additionally, the integrity of the monomeric RBD in F2 was estimated by SDS-PAGE, showing a single band at the expected ~30 kDa in non-reducing and reducing conditions as shown in Figure 1B. Considering these, RBD F2 was employed for the optimization of the anti-SARS-CoV-2 IgG serological tests.

### 3.2. Assay Optimization and Definition of Positive/Negative Absorbance Cutoff Parameters

Fifteen well-characterized RT-PCR positive and 15 RT-PCR negative COVID-19 serum samples were used to optimize the conditions of the S1 and RBD assays. Optimization aimed to maximize the difference between negative and positive serum samples and to establish the positive and negative absorbance cutoff parameters. Each serum sample was processed in two different sets per triplicate by two analysts, for a total of six measurements per tested serum sample.

Figure 2 shows the absorbance values of the S1 and RBD total IgG assays. The S1 total IgG positive sera ranged from 1.42–2.19, and the negative samples ranged from 0.08–0.36. The RBD total IgG assay resulted in an absorbance range of 0.60–1.96 for the positive samples and 0.14–0.38 for the negative samples. Based on these values, we established a cutoff absorbance value of ≥0.50 to report a serum sample as positive for both the S1 and RBD assays. The CV% was <10% in the positive samples, indicating good repeatability of the test as shown in Appendix A.

The cut-off value of ≥0.50 to report a serum sample as positive is three standard deviations above the average value of the negative samples in the RBD, and three standard deviations the average value of the negative samples of the S1 test. For instance, in the last row of Appendix A the average value for the S1 negative serum samples is 0.14, and the standard deviation is 0.07. Hence, 0.14 + (0.07 × 3) = 0.35. Similarly, the RBD negative serum sample average is 0.29, and the standard deviation is 0.07. Hence, 0.29 + (0.07 × 3) = 0.50.

### 3.3. S1 and RDB IgG Initial Validation with 50 COVID-19 RT-PCR Positive Serum Samples

To further validate the assays, we evaluated 50 serum samples from COVID-19 patients confirmed by RT-PCR. Appendix A provides a detailed description of the panel of 50 sera. The results of the assays are shown in Figure 3. Each sample was assayed in three dilutions (1:100, 1:1000, and 1:3000), with the initial dilution (1:100) chosen as working dilution based on the assay optimization protocol defined in the previous section. In the 1:100 dilution, the reactivity of the sera panel against S1 and RBD yielded positive signals in all the samples. Most samples (44/50; 88%) in the S1 test reached absorbance values of >2.00. The remaining six samples (12%) yielded absorbance values between 1.26 and 2.00. The RBD-based assay showed absorbance values of >2.00 in 33 out of the 50 samples (66%), with 16 samples (32%) ranging between 1.14 and 2.00. One sample showed a low but still positive value of 0.56 in the RBD assay. Therefore, both assays detected total IgG anti-SARS-CoV-2 antibodies in 100% of the cases, with the RDB test yielding a slightly lower signal, but higher seropositivity spread. The dilution of 1:1000 showed a wide range of absorbance values from 0.2 to 2.3, and only two samples in the S1 and six samples in the RBD assays were below the positive cut-off determined in the previous section. Similar results were seen in the dilution of 1:3000, in which 4 samples yielded negative results for the S1 protein and 22 samples yielded negative results for RBD. Therefore, we set the 1:100 dilution as required for the correct determination of IgG anti S1 and RBD in our ELISA.

### 3.4. RDB IgG Test Correlation with Age and Sex

Using the RBD-based IgG assay at 1:100 and 1:3000 dilutions of the serum samples, we assessed whether a correlation between anti-RBD IgG and age or sex existed. No correlation or significant difference in anti-RBD IgG was observed with age or sex, respectively, as shown in Appendix A.

### 3.5. Isotyping of the S1 and RBD IgG Antibody Response

We further studied the contribution of each IgG isotype to the antibody response with the panel of 50 sera. The IgG1 isotype predominated the response, with almost all the samples showing a positive response, as presented in Figure 4. The second most intense response was IgG3, whereas low reactivity for IgG2 and IgG4 was detected.

### 3.6. Extended Performance of the RBD-IgG Assay: UDITEST-V2G^®^

Considering that: (i) the RBD assay resolved differences in the IgG titer better than the S1-based test and (ii) the RBD-based assays has been reported to have a good correlation with neutralizing antibodies [8,21,22], we developed a ready-to-use RBD-IgG ELISA kit (UDITEST-V2G^®^) and validated it further Figure 5 shows the first performance test of UDITEST-V2G^®^, which was conducted with samples taken at different points in time after the onset of COVID-9 symptoms. The sera included 887 samples from 198 individuals tested at 7, 14, 21, and 40 days; 204 individuals tested at 7, 14, and 21 days; 214 tested at 7 and 14 days; and 296 individuals who were only tested at day 7 after the onset of symptoms.

A few patients showed detectable IgG titers on day 7. Seroconversion mainly occurred at day 14, with IgG titers showing a significant difference (*p* = 0.0001) at day 14, compared to day 7. This observation is consistent with previous reports [17,23], in which IgG seroconversion in COVID-19 patients develops after 10 days or in the second week after the onset of symptoms. Seventeen out of 198 (8.6%) patients at day 40 post-onset of symptoms did not show IgG titers, as observed in Table 1. This may be due in part to a possible false COVID-19 RT-PCR positive result or a failure of these patients to mount an anti-RBD immune response.

### 3.7. UDITEST-V2G^®^ Independent Performance Test by InDRE

An independent validation of UDITEST-V2G^®^ was performed at InDRE with panel of 378 sera composed of 149 RT-PCR-positive and 229 RT-PCR-negative COVID-19 samples. In this analysis, 148 out of the 149 positive samples were confirmed, whereas 224 out of the 229 negative samples were found negative by UDITEST-V2G^®^. Thus, the clinical sensitivity and clinical specificity of UDITEST-V2G^®^ calculated as: sensitivity = true positive/(true positive + false negative) was 99.33%. The clinical specificity calculated as: specificity = true negative/(true negative + false positive) was 97.82%.

InDRE also evaluated the analytical specificity of UDITEST-V2G^®^ using 28 serum samples from patients with other viral infections as shown in Table 2. All sera were negative according to UDITEST-V2G^®^, to account for 100% specificity.

## 4. Discussion

COVID-19 diagnosis continues to be indispensable in the management of the COVID-19 pandemic, with anti-SARS-CoV-2 IgG serological tests becoming an essential tool to determine individuals who are potentially immune to the viral infection. Here, we described the implementation and validation of a set of SARS-CoV-2 IgG serological assays based on the S1 and RBD proteins as capture reagents. After optimization of the assays to maximize the difference between positive and negative sera, a panel of 50 serum samples from Mexican patients diagnosed as COVID-19 positive by RT-PCR were used to validate the assays. Both assays identified anti-SARS-CoV-2 IgG antibodies in 100% of the serum samples at a 1:100 dilution. The S1-based assay resulted in slightly higher IgG titers than the RBD assay, probably due to the larger molecular size of S1 (~75 kDa) when compared to RBD (~30 kDa) and hence, more exposed immunogenic epitopes in S1 than RBD.

When anti-SARS-CoV-2 IgG isotypes were studied in the serum samples, we found that IgG1 predominated in the IgG response, followed by IgG3. IgG2 and IgG4 were almost nonexistent. Similar results have been reported by Amanat and co-workers [14]. The prevalence of IgG1 and IgG3 titers over IgG2 and IgG4 has also been reported in other viral infections, such as influenza [24]. Moreover, in hepatitis C (HCV) [25], the humoral immune response in 60 patients with chronic HVC and 12 patients acutely infected with HCV showed the prevalence of an IgG1 isotype specific for the HCV antigens. In contrast, the IgG3 isotype was detected in several serum samples, and IgG2 and IgG4 isotypes were rarely or not detected. Furthermore, in hepatitis B infection (HBV) [26], anti-HBs were highly restricted to neutralizing IgG1 and IgG3, with only a minor contribution from IgG2 and IgG4. Additionally, sera from 38 patients with chronic hepatitis B collected 4–10 years after IFN-α therapy [27] showed they responded mainly with IgG1 and/or IgG3.

The human IgG isotypes have the differential capacity to elicit antibody-dependent cellular cytotoxicity (ADCC) (IgG1 and IgG3), antibody-dependent cellular phagocytosis (ADCP) (IgG1, IgG2, IgG3 and IgG4), and complement-dependent cytotoxicity (CDC) (IgG1 and IgG3) [28,29]. Isotype-specific engagement of such immune functions is based on selective Fc receptor interactions on distinct immune cell populations, such as natural killer (NK) cells, neutrophils, and macrophages, as well as C1q. Therefore, it seems that early IgG1 and IgG3 responses to SARS-CoV-2, similar to other acute viral infections, are dominated by isotypes with ADCC and CDC mechanisms. In this case, these findings may be informative in understanding the mechanisms of SARS-CoV-2 infection and disease progression and the decision-making process of isotype engineering to enhance or attenuate effector functions in therapeutic antibody development to treat or prevent COVID-19.

Since the RBD assay showed a better spread of IgG titers, compounded with the reports of good correlation of detection of anti-RBD antibodies with SARS-CoV-2 neutralization functions [17]. The RBD assay was developed as a ready-to-use kit called UDITEST-V2G^®^ and further validated. A first performance test was conducted with a large number of serum samples from COVID-19 patients collected at different times after the onset of COVID-19 symptoms. The results indicated good correspondence between IgG titers and progression of the immune response, in agreement with reports on other serological tests based on ELISA or automated assays [30,31]. A second performance test conducted at InDRE confirmed that UDITEST-V2G^®^ has high sensitivity (99.33%) and specificity (97.82%) with no interference from other viral infections. Therefore, although further evaluations with other coronaviruses are needed to determine whether cross-reactivity may be reported for other serological assays [32], two performance studies conducted by independent laboratories indicate that the UDITEST-V2G^®^ is robust, sensitive, and specific. In fact UDITEST-V2G^®^ has received approval by the sanitary authority in Mexico (COFEPRIS) for commercialization.

In summary, as the COVID-19 pandemic evolves, IgG tests are becoming critical tools to monitor the antibody response against SARS-CoV-2 in recovered patients and vaccinated individuals. We reported here the implementation, optimization, and validation of two ELISA-based assays. Using S1 and RBD as capture reagents and diverse secondary antibodies, we studied a panel of 50 COVID-19 RT-PCR positive sera collected in Mexico City. The IgG isotyping agreed well with previous reports on COVID-19 isotyping studies conducted on other viral infections. The RBD/IgG assay, developed into a ready-to-use kit called UDITEST-V2G^®^, showed to be robust, sensitive and specific leading to approval by the COFEPRIS for commercialization. Worth mentioning is that UDITEST-V2G^®^ is the only 100% Mexican assay approved in Mexico for production, thus highlighting the relevance of this work to assist in the management and eventual control of the COVID-19 pandemic as well as to ascertain the success of COVID-19 vaccination campaigns.

## Figures and Tables

**Figure 1 diagnostics-11-01506-f001:**
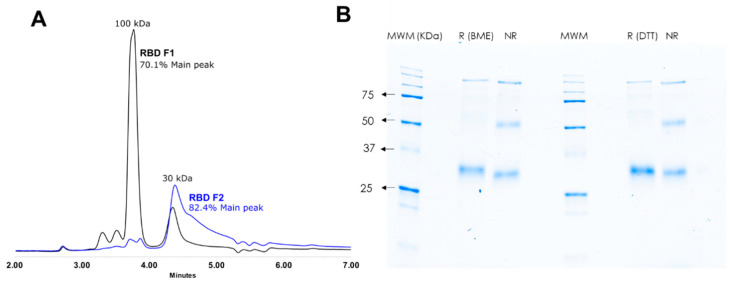
Characterization of the RBD protein. The RBD protein was expressed in HEK 293 cell cultures and purified on a HisTrap™ Nickel column; two fractions were obtained. (**A**) Native SEC analysis evinced that Fraction 1 (RBD F1; black line) was mainly composed of a ~100 kDa protein and Fraction 2 (RBD F2; blue line) by a ~30 kDa protein, which may correspond to the trimer and monomer of the RBD protein, respectively. (**B**) Fraction 2 also exhibits the main band at ~30 kDa when analyzed in denaturing SDS-PAGE using two reducing agents (BME and DTT). Fraction 2 was used for the rest of the work.

**Figure 2 diagnostics-11-01506-f002:**
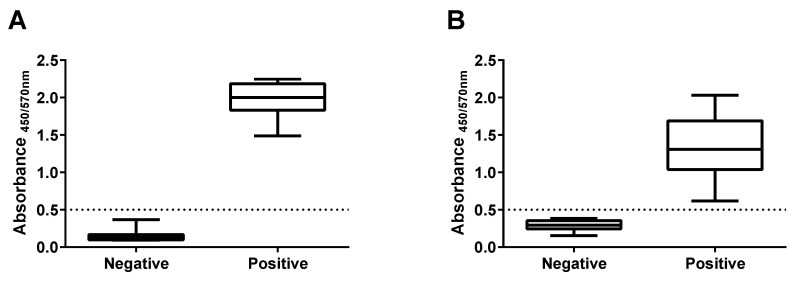
Absorbance values of the 15 negative and 15 positive COVID-19 samples were used to standardize the (**A**) S1- and (**B**) RBD-based IgG assays. Each absorbance value in the plots is the average of six measurements: two sets of triplicates, with one set of triplicates performed one day, and another set of triplicates performed on a different day.

**Figure 3 diagnostics-11-01506-f003:**
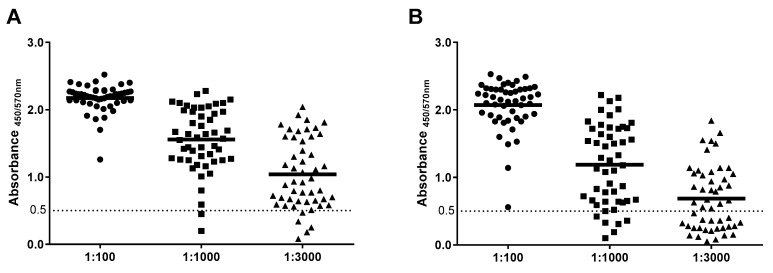
(**A**) S1 and (**B**) RBD test results of the 50 serum samples. Three dilutions (1:100, 1:1000, and 1:3000) per serum sample were assayed. Each data point is the average of duplicates.

**Figure 4 diagnostics-11-01506-f004:**
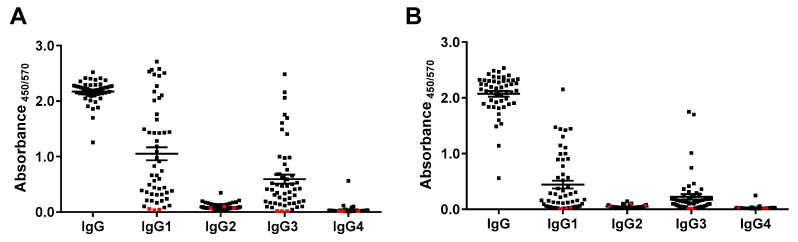
IgG isotyping using (**A**) S1- and (**B**) RBD-based tests. IgG: total IgG. Red squares represent negative controls. All serum samples were assayed at a 1:100 dilution.

**Figure 5 diagnostics-11-01506-f005:**
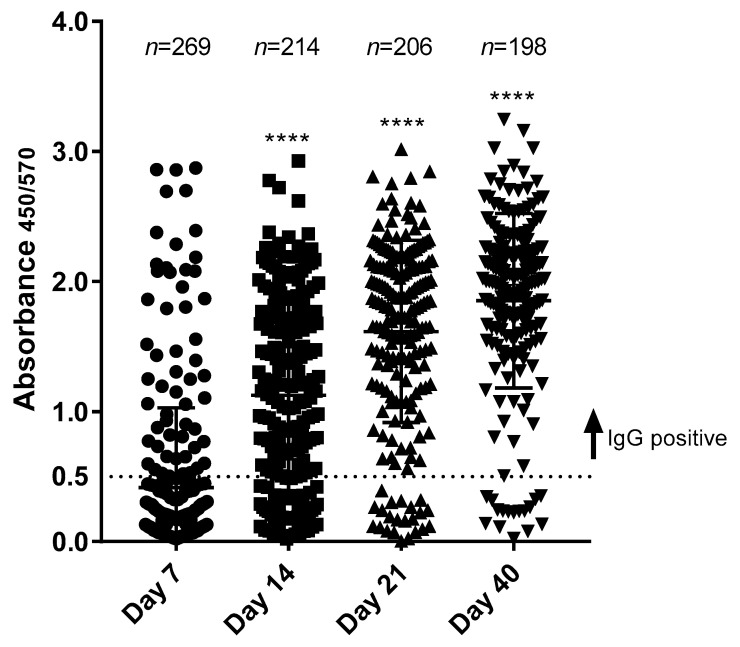
IgG seroconversion of COVID-19 RT-PCR positive patients evaluated with UDITEST-V2G^®^. Serum samples from COVID-19 RT-PCR positive patients were evaluated on day 7, 14, 21, and 40 after the onset of symptoms. Positive and negative IgG samples are shown. **** *p* < 0.0001 student *t*-test.

**Table 1 diagnostics-11-01506-t001:** Number of individuals with positive antibody detection at different days post infection.

	Day 7	Day 14	Day 21	Day 40
Negative Samples	214	56	22	17
Positive Samples	55	158	184	181

**Table 2 diagnostics-11-01506-t002:** The analytical specificity of UDITEST-V2G^®^.

Diagnostic	*n*	IgG	
		Negative	Positive
Hepatitis C	2	2	0
VIH	2	2	0
CMV IgG	1	1	0
TORCH IgM	1	1	0
TORCH IgG	1	1	0
ZIKA IgM	3	3	0
Dengue	7	7	0
CHIK IgM	3	3	0
VIH-Syphilis	4	4	0
Influenza A H1	1	1	0
Influenza A/H1, A/H3, B (Yamagata lineage)	1	1	0
Influenza A/H3	1	1	0
Influenza A/H1, A/H3, B (Victoria lineage)	1	1	0
Total	28	28	0

## Data Availability

All the data used for this work is contained within it or was provided as Appendix A.

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
