# Peer review of "Development and Evaluation of a Set of Spike and Receptor Binding Domain-Based Enzyme-Linked Immunosorbent Assays for SARS-CoV-2 Serological Testing"

_diagnostics, 2021, doi:10.3390/diagnostics11081506_

Round 1

Reviewer 1 Report

This is a well-designed and presented study. I have only some suggestions for further improvement of the manuscript.

  1. The Authors don’t clarify the potential use of S1 and RBD assays for detection or quantitation of anti-SARS-CoV-2. In page 8 [lines 312-313: “Given the robustness of the RBD-IgG test, and because it resolved differences in the 312 IgG titer better than the S1-based test, we developed a ready-to-use RBD-IgG (UDITEST- 313 V2G®). ]. However, based on Table S1 reporting CVs and on Figure 2, the S1 test is more appropriate as qualitative test. On the other hand, as reported in the literature, RBD tests have excellent correlation with neutralizing antibodies. Apparently, the quantitative expression of RBD needs further clarification and comparison with WHO standards and commercially available tests.
  2. Which is the correlation coefficient of absorbance values of S1 and RBD in the various dilutions Figure 3?
  3. Absorbance values of S1 and RBD are higher in patients with severe COVID-19. It would be useful to see this information on Table 2.
  4. In Figure 5, the proportion of positive samples in days 7, 14, 21 and 40 should be reported.
  5. In page 8, line 331 should be written as follows: “Thus, the clinical sensitivity and clinical specificity of UDITEST-V2G
  6. In page 8, line 333 should be written as follows: “The clinical specificity, calculated…”
  7. In page 8, line 335 should be written as follows: “InDRE also evaluated the analytical specificity of UDITEST-V2G® using 28 serum samples from….”

Author Response

Review Report (Round 1)

We much appreciate the comments and suggestions of reviewer #1 to improve the manuscript. A point-by-point response follows.

Comment 1:

The Authors don't clarify the potential use of S1 and RBD assays for detection or quantitation of anti-SARS-CoV-2. In page 8 [lines 312-313: "Given the robustness of the RBD-IgG test, and because it resolved differences in the 312 IgG titer better than the S1-based test, we developed a ready-to-use RBD-IgG (UDITEST- 313 V2G®).]. However, based on Table S1 reporting CVs and on Figure 2, the S1 test is more appropriate as qualitative test. On the other hand, as reported in the literature, RBD tests have excellent correlation with neutralizing antibodies. Apparently, the quantitative expression of RBD needs further clarification and comparison with WHO standards and commercially available tests.

Response:

Lines 71-80 of the manuscript stated:

“S1 and RBD are highly immunogenic and have been extensively utilized as capture reagents for anti-SARS-CoV-2 IgG serodiagnosis [14-18]. Nevertheless, for the most part, current IgG serological tests based on S1 and RBD are rapid qualitative tests and/or are not flexible enough to study different components of the antibody response to SARS-CoV-2 infection; for example, IgG isotypes, anti-SARS-CoV-2 antibody titers, and the immunological competence of a vaccinated individual. To bridge this gap, here we describe the implementation a set of anti-SARS-CoV-2 IgG serological assays based on commercial S1 and in-house produced recombinant RBD as capture reagents.”

In addition, following the referee’s comment, we added in the new version of the manuscript (lines 371-375):

"We developed a commercial ready-to-use RBD-IgG ELISA kit (UDITEST-V2G®) considering i) the robustness of the RBD-IgG system response, ii) that it resolved differences in the IgG titer better than the S1-based test allowing to know the dispersion of specific antibodies without losing positive samples and iii) that it has an excellent correlation with neutralizing antibodies function [8,23,24]."

Regarding the comparison with the WHO standards and commercial tests, we have data showing the concordance of our assays with tests currently available in the market. We are also working on comparison of our positive control with international standard from NIBS. We didn’t include these data in the manuscript due to: (1) space limitations and (2) in our opinion we provide enough information to validate the assays. The data concerning companion with commercial tests and international standards will be part of a second paper we are working on.

Comment 2:

Which is the correlation coefficient of absorbance values of S1 and RBD in the various dilutions Figure 3?

Response:

We used only three serum dilutions in the assays. In some cases, the correlation coefficient of absorbance values of S1 and RBD can be correlated with the dilutions since the absorbance values fall in the dynamic range of the assay. In most of the serum samples however, the first two dilutions (1:100 and 1:1,000) gave similar absorbance values since the titer of antibodies at those dilutions saturate the assay. As expected, diluting these samples further should lead to lower absorbance values and thus a correlation coefficient of absorbance values of S1 and RBD could be calculated. Nevertheless, it should be noted that the goal of the assays was to find a dilution that will identify positive sera and thus higher dilutions of the serum samples will result in a false negative diagnostic.    

Comment 3:

Absorbance values of S1 and RBD are higher in patients with severe COVID-19. It would be useful to see this information on Table 2.

Response:

The goal of this manuscript was to implement and validate ELISA tests to identify anti-SARS-CoV2 IgG antibodies. To that end, the samples used in the validation were collected from individuals with moderate or mild COVID-19 symptoms. A study of patients with severe COVID-19 would certainly be information but it is beyond the scope of this study.  

Comment 4:

In Figure 5, the proportion of positive samples in days 7, 14, 21 and 40 should be reported.

Response:

We included in the revised version of the manuscript a table as part of Figure 5 showing the proportion of positive and negative samples as follows:

Day 7

Day 14

Day 21

Day 40

Negative

214

56

22

17

Positive

55

158

184

181

Total sample

269

214

206

198

Minor comments:

  1. In page 8, line 331 should be written as follows: "Thus, the clinical sensitivity

and clinical specificity of UDITEST-V2G

Change included in the revised version.

  1. In page 8, line 333 should be written as follows: "The clinical specificity, calculated…"

Change included in the revised version.

  1. In page 8, line 335 should be written as follows: "InDRE also evaluated the analytical specificity of UDITEST-V2G® using 28 serum samples from…."

Change included in the revised version.

Reviewer 2 Report

  1. Please indicate why you intend to develop ELISA to evaluate the spike protein of SARS-CoV-2 and the significance using ELISA to instead or improve RT-PCR in the “Introduction” section.
  2. Please provide the precision, limit of detection (LOD) and limit of quantification (LOQ) for the detection and evaluation of the spike protein of SARS-CoV-2 in the “Result” section.
  3. Please add a paragraph in the “Discussion” section to discuss the advantages using ELISA over RT-PCR to evaluate the spike protein of SARS-CoV-2.

Author Response

Review Report (Round 1)

Comment 1:

Please indicate why you intend to develop ELISA to evaluate the spike protein of SARS-CoV-2 and the significance using ELISA to instead or improve RT-PCR in the “Introduction” section.

Response:

Please note that the work reported in the manuscript aimed to implement and validate an ELISA to identify anti-SARS-CoV2 IgG antibodies using S1 or RBD as a capture reagent. We are not quantifying the spike protein of SARS-CoV-2 in serum samples. The relevance of or work is stated in discussion as follows (lines 405 –410):

 “As the COVID-19 pandemic evolves, IgG tests are becoming critical tools to monitor the antibody response against SARS-CoV-2 in recovered patients and vaccinated individuals, and UDISTEST-V2G® is the only 100% Mexican assay approved in Mexico for production, thus highlighting the relevance of this work to assist in the management and eventual control of the COVID-19 pandemic, as well as to ascertain the success of COVID-19 vaccination campaigns”.

Comment 2:

Please provide the precision, limit of detection (LOD) and limit of quantification (LOQ) for the detection and evaluation of the spike protein of SARS-CoV-2 in the “Result” section.

Response:

We didn’t develop a test to measure S1 nor RBD. See response to comment 1.

Comment 3:

Please add a paragraph in the “Discussion” section to discuss the advantages using ELISA over RT-PCR to evaluate the spike protein of SARS-CoV-2.

Response:

We didn’t develop a test to measure S1 nor RBD. See response to comment 1.

Round 2

Reviewer 2 Report

No.

This manuscript is a resubmission of an earlier submission. The following is a list of the peer review reports and author responses from that submission.

Round 1

Reviewer 1 Report

  1. This paper claims to be the first such study on this subject area from Mexico and therefore is of interest. However, the sample numbers are relatively small and the novelty of the approach is limited as many similar test formats have been reported already.
  2. Overall the usage of English needs to be edited and improved.
  3. In Figure 1 the legend needs to be more explanatory clearly setting out what F1 and F2 mean. In addition, full details on the experimental approach used needs to be included in the relevant section of the paper.
  4. Suggest use Absorbance rather than OD
  5. Need to check the number of significant figures given. Some appear incorrect relative to others
  6. It is not clear how the SDs were determined.
  7. The exact way in which the results were treated and analysed need to be clearer. Were each of the sera analysed separately on at least 3 different occasions?

Author Response

Responses to referee 1.

We much appreciate your comments and suggestions to the manuscript, which have been very helpful to improve the quality of the text, presentation of the results and discussion of our findings. A point-by-point response follows.

  1. This paper claims to be the first such study on this subject area from Mexico and therefore is of interest. However, the sample numbers are relatively small and the novelty of the approach is limited as many similar test formats have been reported already.

Response 1.

Please note that we have rewritten the manuscript in its entirety, included new data sets, added new information pertaining the validation of the assays and discussed the relevance of the tests implemented and validated in the manuscript.  

Specifically:  

  1. Included the study a new panel of 887 serum samples collected from positive COVID-19 RT-PCR individuals at 4 collection times (7, 14, 21, 40 days) after the onset of COVID-19 symptoms. This study significantly expanded the number of serum samples in the old version of the manuscript (50 serum samples) to more than ten times the number of samples in the new version of the manuscript. The results provide more reliable information on the seroconversion times of patients with COVID-19 in the Mexican population than the original manuscript.
  1. Included a performance study done in collaboration with Instituto de Diagnóstico y Referencia Epidemiologicos (InDRE), which is the reference organization in Mexico for the diagnostics of COVID-19. This performance test was conducted with a total of 406 serum samples. This study provided an assessment by a third part of the sensitivity and specificity of the assay.
  1. Removed redundant information such as the seroconversion study with a small panel of serum samples.
  1. Included a comment on the approval for commercialization by the regulatory agency in Mexico (COFEPRIS) of one of the assays. This is the only 100% Mexican assay approved in Mexico for commercialization, which speaks to the relevance of the work reported in the manuscript.

  1. Overall, the usage of English needs to be edited and improved.

Response 2.

The new version of the manuscript has extensively been reviewed to correct typos and improve the style of the manuscript, including a detailed review and correction of the use of English language.

  1. In Figure 1 the legend needs to be more explanatory clearly setting out what F1 and F2 mean. In addition, full details on the experimental approach used need to be included in the relevant section of the paper.

Response 3.

Updated the footnote of Figure 1 and included a more extensive explanation of the RBD purification and characterization processes.

  1. Suggest use Absorbance rather than OD

Response 4.

Changed OD to Absorbance in the manuscript and Figures.

  1. Need to check the number of significant figures given. Some appear incorrect relative to others.

Response 5.

Figure and Table numbering have been reviewed and corrected as needed.

  1. It is not clear how the SDs were determined.

Response 6.

A detailed description of the standard deviation (SD) calculation is now included in the manuscript.

  1. The exact way in which the results were treated and analyzed need to be clearer. Were each of the sera analyzed separately on at least 3 different occasions?

Response 7.

The manuscript has been rewritten in its entirety to make it clear the source of the samples and the treatments applied to each sera panel.

Reviewer 2 Report

this paper describes a new diagnostic set for SARScov 2 infection.

The paper needs to be edited, before publication

in the background,  epidemiological data need to be improved with recent data.

The paragraph "nonetherless 35% false negative........ during days 15to39" is not entirely correct, the performance of the recent RTPCR is superior and the citations are for antibodies not for PCR.

In the discussion The first paragraph is incorrect (see 10.12998/wjcc.v8.i19.4280, 10.1002/iid3.400) and could be replaced by diagnostic difficulties at the first visit (see 10.1002/iid3.440).

Author Response

Responses to referee 2.

We much appreciate your comments and suggestions to the manuscript, which have been very helpful to improve the quality of the text, presentation of the results and discussion of our findings. A point-by-point response follows.

  1. this paper describes a new diagnostic set for SARScov 2 infection.

Responses 1.

We truly appreciate and encouragement of the referee’s comment.

  1. The paper needs to be edited, before publication.

Responses 2.

The manuscript has been rewritten in its entirety improve the style, and presentation and discission of the results.

  1. in the background, epidemiological data need to be improved with recent data.

Responses 3.

The background, epidemiological data and references have been updated and edited to provide a better context for the results and discussion of the manuscript.

  1. The paragraph "nonetheless 35% false negative........ during days 15to39" is not entirely correct, the performance of the recent RTPCR is superior and the citations are for antibodies not for PCR.

Responses 4.

This paragraph was removed from the manuscript.

  1. In the discussion The first paragraph is incorrect (see 10.12998/wjcc.v8.i19.4280, 10.1002/iid3.400) and could be replaced by diagnostic difficulties at the first visit (see 10.1002/iid3.440).

Responses 5.

This paragraph was removed from the manuscript and the references updated.
